# Endovascular baroreflex amplification and the effect on sympathetic nerve activity in patients with resistant hypertension: A proof-of-principle study

**Monique E. A. M. van Kleef**[1], **Karsten Heusser**[2], **André Diedrich**[3], **P. Liam Oey**[4], **Jens Tank**[2], **Jens Jordan**[2], **Peter J. Blankestijn**[4], **Bryan Williams**[5], **Wilko Spiering**[1]*

1 Department of Vascular Medicine, University Medical Center Utrecht, Utrecht, The Netherlands, 2 Institute of Aerospace Medicine, German Aerospace Center, Cologne, Germany, 3 Division of Clinical Pharmacology, Department of Medicine, Autonomic Dysfunction Center, Vanderbilt University Medical Center, Nashville, TN, United States of America, 4 Department of Nephrology, University Medical Center Utrecht, Utrecht, The Netherlands, 5 University College London (UCL) Institute of Cardiovascular Science and National Institute for Health Research (NIHR) UCL Hospitals Biomedical Research Centre, London, United Kingdom

* W.Spiering@umcutrecht.nl

**Data Availability Statement:** The data underlying the findings of the present study are contained within the files attached to this submission.

## Abstract

### Background

First in human studies suggest that endovascular baroreflex amplification (EVBA) lowers blood pressure (BP). To explore potential mechanisms for BP reduction, this study examines the effects of EVBA on muscle sympathetic nerve activity (MSNA) and baroreceptor sensitivity (BRS).

### Methods

In a single-center sub-study of the CALM-DIEM study (Controlling And Lowering blood pressure with the MobiusHD—Defining Efficacy Markers), 14 patients with resistant hypertension were treated with EVBA. Microneurography and non-invasive continuous BP measurements were performed at baseline and three months after MobiusHD implantation. The primary outcome was change in MSNA. Secondary outcomes were change in baroreflex sensitivity (BRS), cardiovascular responses to a sympathetic stimulus, BP, heart rate (HR) and heart rate variability (HRV).

### Results

The primary endpoint was obtained in 10 of 14 patients enrolled in the sub-study. MSNA burst frequency and burst incidence decreased in 6 of 10 patients: mean change -4.1 bursts/min (95% confidence interval -12.2 to 4.0) and -3.8 bursts/100 heartbeats (-15.2 to 7.7). MSNA spike frequency and spike count decreased in 8 of 10 patients: mean change -2.8 spikes/sec (-7.3 to 1.8) and -3.0 spikes/heartbeat (-6.1 to 0.1). Change in MSNA and BP were not correlated. Office BP decreased by -14/-6 mmHg (-27 to -2/-15 to 3). We observed a trend towards decreased HR (-5 bpm, -10 to 1) and increased total power HRV

**Funding:** The study was funded by the manufacturer of the MobiusHD implant, Vascular Dynamics, Inc. André Diedrich is supported by the National Heart, Lung, and Blood Institute of the National Health (Award Number NIH 1R56HL142583-01, 1R01HL142583-01). Bryan Williams is supported by the NIHR University College London Hospitals Biomedical Research Centre. Vascular Dynamics, Inc. was involved in the design of the main study, data monitoring and central storage of study data. The funder had no role in data analysis or preparation of the manuscript which was directed by the manuscript authors. Monique E.A.M. van Kleef and Wilko Spiering had access to all study data and had final responsibility for the decision to submit the paper for publication. The National Heart, Lung, and Blood Institute of the National Health and the NIHR University College London Hospitals Biomedical Research Centre had no role in study design, data collection and analysis, decision to publish, or preparation of the manuscript.

**Competing interests:** Monique E.A.M. van Kleef was indirectly paid from a research grant by Vascular Dynamics, Inc. Karsten Heusser, P. Liam Oey, Jens Tank, André Diedrich and Peter J. Blankestijn report no conflict of interest. Jens Jordan served as consultant for Novartis, Boehringer-Ingelheim, Sanofi, Orexigen, Riemser, Vivus, and is cofounder of Eternygen GmbH. Bryan Williams has received honoraria for consultancy from Vascular Dynamics Inc. W. Spiering is a consultant for Vascular Dynamics and has received a research grant from Vascular Dynamics. This does not alter our adherence to PLOS ONE policies on sharing data and materials.

(623 msec$^2$, 78 to 1168). In contrast, BRS and cardiovascular responses remained unchanged after EVBA.

## Conclusions

In this proof-of-principle study, EVBA did not significantly decrease MSNA in patients with resistant hypertension. EVBA did not impair baroreflex function.

## Trial registration

Clinical trial registration at NCT02827032.

## Introduction

Sympathetic overdrive is regarded as one of the key characteristics in patients with primary hypertension and is exceptionally high in patients with resistant hypertension [1]. Therefore, several treatment strategies have focused on decreasing sympathetic activity in order to lower blood pressure (BP). An exceptional target is the carotid baroreceptor which undergoes complex changes in hypertensive patients [2] and plays a central role in the development and sustainment of the hypertensive continuum.

Electrical baroreflex activation was the first antihypertensive device therapy targeting the carotid baroreflex applied in humans and showed a decrease in sympathetic activity and BP [3–5]. However, electrical pulse generator implantation is an invasive surgical procedure under general anesthesia [6], it is associated with unfavourable side effects such as jaw or neck pain, nerve injury, globus sensation, cough or dysphonia, and requires battery replacement every few years [7]. Therefore, a less invasive procedure, also relying on baroreflex modulation, has been developed: endovascular baroreflex amplification (EVBA) by the implantable MobiusHD device. The MobiusHD is an endovascular device implanted in the carotid sinus. It is assumed to reshape the carotid artery intermittently during the cardiac cycle, thereby increasing vessel wall strain and baroreceptor firing. It is hypothesized that this triggers the baroreflex pathway leading to decreased sympathetic activity and a fall in BP [8, 9].

The first-in-human study of EVBA showed a favourable and sustainable effect on BP with an acceptable safety profile six months after implantation [8]. However, the mechanism by which EVBA reduces BP has never been studied. It is also unknown whether and to what extent the device interferes with baroreflex function. Therefore, the primary aim of this proof-of-concept study was to determine the effect of EVBA on muscle sympathetic nerve activity (MSNA). Secondary aims were to determine if the MobiusHD device changes baroreceptor sensitivity (BRS) and sympathetic cardiovascular reactivity. Additionally, we studied the effect of EVBA on BP, heart rate (HR) and heart rate variability (HRV). Finally, we present the 3-month safety.

## Methods

### Study design and population

Patients with resistant hypertension who attended the outpatient hypertension clinic of the University Medical Center Utrecht were asked to participate in this single-centre sub-study of the CALM-DIEM study (Controlling And Lowering blood pressure with the MobiusHD—Defining Efficacy Markers) by their physician. The CALM-DIEM study was an observational

study including patients (18–80 years of age) with primary resistant hypertension (24-h mean ambulatory systolic BP >130 mmHg on a stable regimen of at least three antihypertensive medications, including a diuretic). Fourteen patients were included between November 2016 and November 2018. Main exclusion criteria were: hypertension secondary to an identifiable and treatable cause other than sleep apnea; any plaque, ulceration or stenosis in the carotid artery or the aortic arch; and carotid artery lumen <5.00 mm or >11.75 mm or too much tapering at the planned location for implantation (S1 File). Patients in the sub-study consented to undergo additional measures in our hospital. The study was approved by Medical Ethics Committee United (Nieuwegein, the Netherlands) and conforms to the 1975 Declaration of Helsinki ethical guidelines. All patients gave their written informed consent. Safety was monitored by an independent Data Safety and Monitoring Board (DSMB). The CALM-DIEM study is registered at www.clinicaltrials.gov (NCT02827032). Because of its small size and the enrolment of patients in a single center only, the present sub-study was initially not registered with the main study. The sub-study was added after enrolment of the participants started. We confirm that all ongoing and related trials for this intervention are registered.

## Study procedures

All patients underwent carotid duplex and computed tomography angiography to rule out carotid anatomical anomalies, plaque, or stenosis that would preclude MobiusHD placement. After confirmation of eligibility, antihypertensive medications were discontinued for a period of 2 weeks (beta blockers required a preceding 2-week tapering scheme) before sympathetic and cardiovascular measurements were performed. If deemed appropriate, i.e. when escape medication was required during medication washout in the prior diagnostic programme, calcium channel blockers were prescribed, with a similar dose at baseline and 3 months. This strategy was implemented to ensure that antihypertensive medication use remained stable over time.

MobiusHD implantation was preceded by bilateral carotid angiography to examine whether the internal carotid arteries were anatomically suitable for implantation and to determine implant size: 5.00–7.00 mm (A), 6.25–9.00 mm (B) and 8.00–11.75 mm (C). The MobiusHD was implanted unilaterally on the anatomically best suited side. The implantations were performed by an experienced neurointerventionist who had performed sixteen MobiusHD implantations before and more than hundred conventional carotid stent implantations. The procedure has been described in detail before [8]. Intravenous heparin was administered during the implantation procedure. All patients received dual antiplatelet therapy with 80 mg aspirin and 5–10 mg prasugrel (or equivalent) from 5 days before implantation to 3 months after implantation. Aspirin was continued indefinitely.

## Sympathetic and cardiovascular measurements

Sympathetic and cardiovascular measurements were performed at baseline and 3 months after MobiusHD implantation after a light breakfast in the morning hours. Patients had to empty their bladder just before start of the measurements and were discouraged from engaging in vigorous exercise, smoking and drinking alcohol, coffee, tea or other beverages containing caffeine in the 24-h before the measurements. Measurements were performed in supine position in a dimly-lit room. Non-invasive beat-by-beat BP and ECG were recorded continuously (Finapres NOVA, FMS, Amsterdam, the Netherlands). In addition, brachial artery BP was measured twice with appropriately sized cuffs, on the left arm and 1 min apart. Sympathetic nerve activity was recorded with microelectrodes (UNP35FAT or UNP35F2S, FHC Inc., Bowdoin, United States, after 10 cases we switched to UNP35F2S to improve quality) from the

right peroneal nerve using ADInstruments (Powerlab 16/35 with Neuro Amp EX 4, NSW, Australia) equipment. Microneurography was performed by an experienced neurophysiologist (L.O.) who had performed hundreds of procedures previously [10, 11]. The MSNA signal was identified by a classical response to the Valsalva maneuver and absence of response to tactile and auditory stimulation. After instrumentation and detection of the nerve signal, sympathetic and cardiovascular parameters were measured under the following conditions: 1) twice, each for 5 min, in resting supine position; 2) during two 15-sec 40 mmHg Valsalva maneuvers (ANS application of the Finapres NOVA system); and 3) during a 2-min cold pressor test (S1 Fig). At baseline and 3 months, ambulatory BP was recorded by an oscillometric device (SpaceLabs 90217, Spacelabs Healthcare, Issaqua, WA, United States) on the non-dominant arm, before antihypertensive medication washout. BP was recorded every 20 min during the day and every 60 min during the night.

### Resting muscle sympathetic nerve activity

Data were analog-to-digital converted and analyzed using a program written by one of the authors (A.D.). The analyst (K.H.) was blinded for the patients and timepoints. The following parameters were determined from the integrated nerve signal obtained during the two 5-min resting state measurements: number of bursts per min (burst frequency) and number of bursts per 100 heartbeats (burst incidence). In addition to the planned burst analysis of the integrated signal, we also determined the effect of EVBA on raw MSNA action potential content parameters: mean beat-to-beat median spike frequency and mean beat-to-beat spike count [12, 13] which can reveal additional physiological information about the firing pattern that may get lost in the integration process for the common envelope curve of integrated MSNA burst activity. All four parameters were averaged over the two 5-min resting episodes. Patients with frequent premature beats (>10% of the beats) or insufficient quality of the MSNA signal (signal-to-noise ratio ≤2) were excluded from analysis.

### Cardiac baroreflex sensitivity

Spontaneous cardiac BRS was assessed by using the sequence technique to calculate the slope of the linear regression line between systolic BP and subsequent RR-intervals (within the same or the next heartbeat) [14, 15]. Sequences with at least three intervals, 0.5 mmHg systolic BP change, and 0.5 msec RR-interval change, were analyzed only if the R-Squared values were >0.85. The lag between the systolic BP and R-top which produced the maximum number of significant sequences (0, 1 or 2 beats) was selected. The median slope of significant sequences was determined and averaged over the two 5-min resting periods. In addition, spontaneous cardiac BRS was calculated by the cross-correlation method (xBRS) implemented in the Finapres software. In short, xBRS computes the correlation between systolic BP and RR-interval, resampled at 1 Hz, in a sliding 10-second window, with delays of 0–5 seconds for interval [16]. The delay with the greatest positive correlation is selected and, when significant at p<0.01, the slope is recorded as one xBRS value. The xBRS values obtained during 0:10–5:00 min of the two resting state measurements were averaged. Evoked cardiac BRS was determined by estimating the linear regression slope of RR-interval plotted against systolic BP in early phase II of the Valsalva maneuver (S2 Fig), only if the correlation coefficients were >0.85. The average slope of two Valsalva maneuvers was calculated. Patients with arrhythmias (i.e. frequent premature beats) or less than three significant sequences were excluded from analysis.

## Sympathetic baroreflex sensitivity

Spontaneous sympathetic BRS was assessed in the two 5-min resting recordings using the threshold technique as described by Kienbaum et al [17]. Diastolic BPs were grouped into bins of one mmHg and within these bins the percentage of heartbeats associated with a burst was determined. Linear regression analysis was performed to determine the threshold slope of the percentages plotted against the means of the diastolic BP bins, weighted for the number of cardiac cycles within each bin [18]. A p-value of <0.05 was used as the criterion to accept the slope. Additionally the $T_{50}$ was determined, which is the diastolic BP at which 50% of the heartbeats is associated with a burst. The threshold slopes and $T_{50}$'s were averaged over the two 5-min resting episodes. Evoked sympathetic BRS was assessed by dividing the change in median spike rate by the change in systolic BP during early phase II of the Valsalva maneuver (S2 Fig), compared to the 15 seconds before the Valsalva maneuver [19]. The average of two Valsalva's was taken. Patients with frequent premature beats, insufficient quality of the MSNA signal or non-significant threshold slopes (for evoked sympathetic BRS only) were excluded from analysis.

## Cardiovascular reactivity to sympathetic stimulus

Cardiovascular reactivity to a sympathetic stimulus was assessed by examining the response to the cold pressor test. Patients were asked to submerge their hand (up to the wrist) in a bucket of ice water, for 2 min. MSNA burst indices, mean arterial BP and heart rate during the second min of the cold pressor test were compared to the 2-min pre-cold pressor test averages and the change was recorded [20–22]. As for the estimation of resting MSNA, patients with frequent premature beats or insufficient quality of the MSNA signal during the cold pressor test were excluded from analysis of MSNA indices.

## Heart rate variability

Power spectral density of heart rate variability (HRV) was estimated from the two 5-min recordings of RR-intervals during spontaneous breathing with the fast Fourier transformation-based Welch algorithm [19]. Total power (TP), power in the low-frequency range (LF, 0.04–0.15 Hz), high frequency range (HF, 0.15–0.40 Hz) and the LF to HF ratio (LF:HF) were calculated according to the Task Force recommendations [23]. Patients with frequent premature beats or arrhythmias were excluded from analysis.

## Safety outcomes

Safety outcomes, as defined in the main study, were: 30-day major adverse clinical events (death, stroke and/or myocardial infarction), periprocedural device-related serious events (carotid artery rupture, dissection, aneurysm, stenosis or occlusion), serious adverse events and unanticipated adverse device effects.

## Statistical analysis

Baseline characteristics are presented as mean ± SD or median ± IQR. Mean sympathetic and cardiovascular parameters presented in the remaining tables are presented with their SE. Mean changes are presented with corresponding 95% confidence interval (95% CI). Change in MSNA, BRS, cardiovascular reactivity to sympathetic stimulus, BP, HR and HRV were tested by the paired t-test. The relationship between change in MSNA and change in office BP was assessed by Pearson correlation analysis. Adverse events were reported as counts and percentage of the total study population. Although a sample size calculation is uncommon in a proof-

of-principle study, we estimated that a minimum of 9 paired MSNA measurements was needed to show a change of 10 bursts/min or 15 bursts/100hb with 90% power (two-sided tests at the 5% level). Enrollment stopped after these 9 pairs were successfully obtained. A p-value of <0.05 was considered statistically significant. All statistical analyses were performed with R, version 3.5.1 (R Development Core Team, Vienna, Austria).

### Role of the funding source

The study sponsor (Vascular Dynamics, Inc.) was involved in study design of the main study, data monitoring and central storage of the data. The study sponsor was not involved in design of the sub-study and had no role in data analysis, data interpretation or manuscript preparation. The authors had full access to the study data and had final responsibility for the decision to submit the paper for publication.

## Results

Forty-two patients were screened for eligibility (Fig 1), 14 (mean age 51.6 (± 6.8) years, four women) were implanted with the MobiusHD device, 8 (57%) on the right and 6 (43%) on the left (Table 1). One patient was unable to attend the 3-month follow-up visit due to psychosocial issues, which were absent at time of inclusion, leaving 13 patients available for analysis of sympathetic and cardiovascular parameters. An overview of the number of patients in- and excluded per parameter and the reasons for exclusion is provided in S1 Table.

### Resting muscle sympathetic nerve activity

Burst frequency and burst incidence decreased in 6 of the 10 patients (60%) (Fig 2 and S2 Table), mean beat-to-beat median spike frequency and mean beat-to-beat spike count

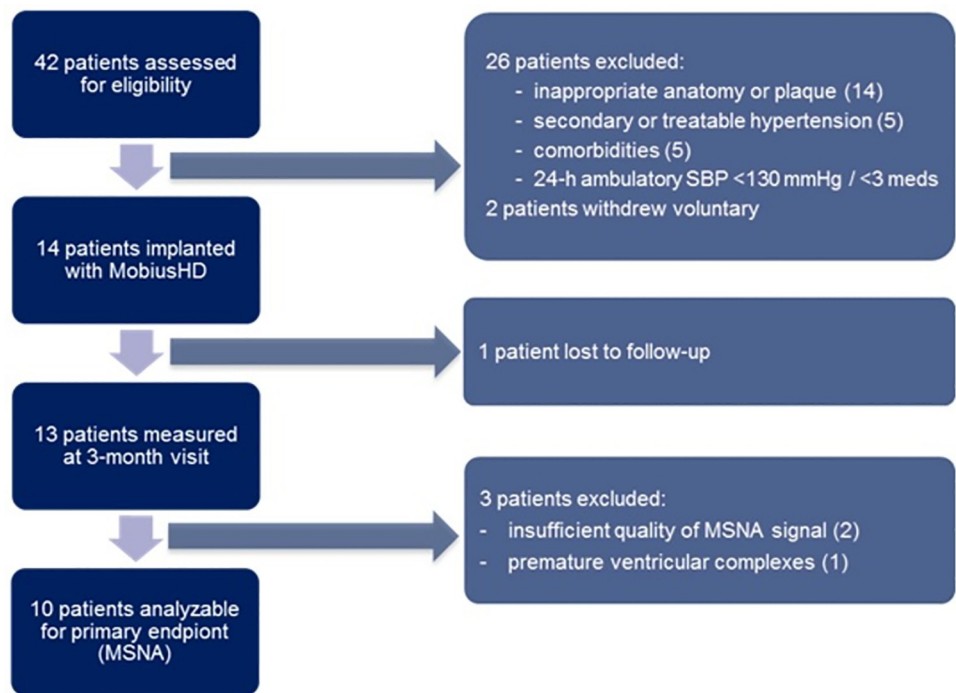

**Fig 1. Flow chart of the patients screened, excluded, lost to follow-up and excluded from analysis of the primary endpoint.** SBP = systolic blood pressure, meds = antihypertensive medications, MSNA = muscle sympathetic nerve activity.

**Table 1. Baseline characteristics of all implanted patients.**

| | Patients implanted |
|---|---|
| | (n = 14) |
| Age (years) | 51.6 (6.8) |
| Women | 4 (29%) |
| Diabetes mellitus | 2 (14%) |
| Cardiovascular disease | 1 (7%) |
| Prior renal denervation | 4 (29%) |
| Obstructive sleep apnea syndrome | 4 (29%) |
| Current smoker | 2 (14%) |
| Body mass index (kg/m$^2$) | 27.9 (4.9) |
| eGFR (ml/min/1.73m$^2$) | 81 (12) |
| Office blood pressure (mmHg) | 178/107 (25/17) |
| Heart rate (bpm) | 79 (10) |
| 24-h ambulatory blood pressure (mmHg) | 156/95 (20/15) |
| Number of antihypertensive medications | 3 (3–4) |
| Daily defined dose | 5 (3–5) |
| A+C+D regimen | 11 (79%) |
| A+C+D+MRA regimen | 2 (14%) |
| Escape medication | 9 (64%) |
| Side of MobiusHD implantation (left) | 6 (43%) |

Data displayed as mean (SD), median (IQR) or n (%). eGFR estimated by the CKD-EPI formula. A = ACE-inhibitor, angiotensine-II-receptor blocker or direct renin inhibitor, C = calcium channel antagonist, D = diuretic, MRA = mineralocorticoid receptor antagonist.

decreased in 8 of the 10 patients (80%). Point estimates of change were: -4.1 bursts/min (95% CI -12.2 to 4.0) (burst frequency), -3.8 bursts/100 heartbeats (-15.2 to 7.7) (burst incidence), -2.8 spikes/sec (-7.3 to 1.8) (beat-to-beat median spike frequency) and -3.0 spikes/beat (-6.1 to 0.1) (beat-to-beat spike count) (Table 2). The trend to decrease in MSNA could not be explained by change in quality of the MSNA measurement, since MSNA signal-to-noise ratio at baseline and 3 months post-implantation were similar. Change in MSNA and change in office systolic BP were not correlated (Fig 2).

## Baroreflex sensitivity and cardiovascular responses to sympathetic stimulus

Sympathetic and cardiac BRS were not affected by EVBA (Table 2). In addition, MSNA, MAP and HR response to cold pressor testing remained unchanged following EVBA.

## Blood pressure, heart rate and heart rate variability

Mean office BP (measured after medication washout on the day of MSNA recording) decreased by -14/-6 mmHg from baseline (95% CI -27 to -2/-15 to 3) (Table 2). Mean 24-h ambulatory BP (measured while patients were on medication) did not decrease significantly (-3/-4 mmHg, 95% CI -13 to 7/-12 to 4). HR decreased in 9 of 13 patients (69%), however, also this decrease was not significant: -5 bpm (95% CI -10 to 1) on average. There was a trend to increased mean HRV in the total power (623 msec$^2$, 95% CI 78 to 1168) and high frequency spectral power (59 msec$^2$, 95% CI -1.5 to 120) (Table 2).

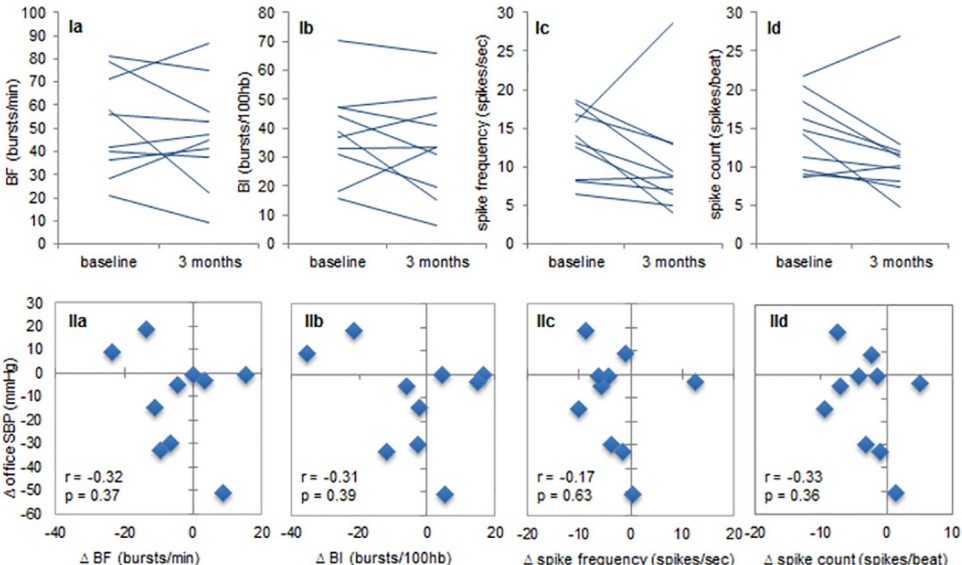

**Fig 2. Individual MSNA changes and correlation with change in BP.** Panel I: individual changes in burst frequency (Ia), burst incidence (Ib), mean beat-to-beat median spike frequency (Ic), and mean beat-to-beat spike count (Id). Panel II: correlation between change in office SBP and change in burst frequency (Ia), burst incidence (IIb), mean beat-to-beat median spike frequency (IIc), and mean beat-to-beat spike count (IId). Observations in the left lower quadrants of the plots support baroreflex-mediated BP reduction. SBP = systolic blood pressure, BF = burst frequency, BI = burst incidence, r = Pearson's correlation coefficient, p = p-value.

## Safety

One major adverse clinical event occurred in one patient (7%) after MobiusHD implantation on the left (S3 Table). This patient developed vertigo without light headedness on the day after the procedure. Diffusion-weighted MRI, which was performed eight days post-procedure, showed diffusion restriction in the right putamen indicating recent ischemia. The DSMB attributed this event to extreme hypotenstion that occurred post-procedure: office systolic BP dropped from 189/112 mmHg at baseline to approximately 100/50 mmHg in the days after the procedure. We did not observe any periprocedural device-related serious events. Two patients (14%) developed post-procedural groin bleeding which required intervention by means of a femostop. Both patients remained hemodynamically stable and did not require blood transfusions. In one patient this lead to prolonged hospitalization and was therefore categorized as a serious adverse event. Other adverse events that were related or possibly related to the device or procedure were: temporary decrease in kidney function (eGFR decrease of >10%, n = 3, 21%), pain at the puncture site (n = 2, 14%), dizziness (n = 1, 7%), epistaxis and axillary hematoma (n = 1, 7%), headache (n = 1, 7%), and periprocedural chest pain without typical electrocardiographic changes or elevated mycardial enzymes (n = 1, 7%). We did not observe any unanticipated adverse device effects.

## Discussion

In this proof-of-principle study EVBA did not result in a significant decrease in MSNA in patients with resistant hypertension. Furthermore, EVBA did not impair cardiovascular compensatory mechanisms.

Since sympathetic overactivity is one of the key features of hypertension, EVBA aims to reduce sympathetic activity through the negative feedback mechanism of the baroreflex. In the present study we quantified systemic sympathetic activity by measuring MSNA from the

**Table 2. Sympathetic and cardiovascular parameters at baseline and 3 months after implantation.**

| | N | baseline | 3 months | p-value |
|---|---|---|---|---|
| **MSNA** | | | | |
| Burst frequency (bursts/min) | 10 | 38.3 (5.0) | 34.2 (5.6) | 0.284 |
| Burst incidence (bursts/100hb) | 10 | 51.2 (6.7) | 47.5 (7.2) | 0.476 |
| Mean beat-to-beat median spike freq (spikes/sec) | 10 | 13.2 (1.4) | 10.4 (2.2) | 0.200 |
| Mean beat-to-beat spike count (spikes/beat) | 10 | 13.5 (1.5) | 11.9 (1.9) | 0.059 |
| **BRS** | | | | |
| cBRS Valsalva (msec/mmHg) | 12 | 2.7 (0.5) | 2.6 (0.4) | 0.738 |
| cBRS Sequence up (msec/mmHg) | 10 | 4.5 (0.9) | 4.3 (0.5) | 0.774 |
| cBRS Sequence down (msec/mmHg) | 10 | 4.8 (0.8) | 5.6 (0.7) | 0.106 |
| xBRS (msec/mmHg) | 10 | 5.0 (0.6) | 5.7 (0.9) | 0.325 |
| sBRS Valsalva (spikes/sec/mmHg) | 8 | -0.6 (0.1) | -0.5 (0.2) | 0.621 |
| sBRS $T_{50}$ (mmHg) | 9 | 79.0 (6.5) | 76.0 (8.1) | 0.516 |
| sBRS treshold slope (%/mmHg) | 9 | -2.3 (0.3) | -2.3 (0.3) | 0.884 |
| **CPT response** | | | | |
| Δ MAP to CPT (mmHg) | 13 | 34.6 (4.2) | 33.9 (3.8) | 0.855 |
| Δ HR to CPT (bpm) | 13 | 11.7 (1.9) | 11.1 (1.9) | 0.631 |
| Δ burst incididence to CPT (bursts/100hb) | 8 | 16.8 (5.3) | 19.1 (6.2) | 0.567 |
| Δ burst frequency to CPT (bursts/min) | 8 | 17.8 (5.1) | 20.3 (5.5) | 0.484 |
| **BP and HR** | | | | |
| Office SBP (mmHg) | 13 | 182 (6) | 168 (6) | 0.027 |
| Office DBP (mmHg) | 13 | 112 (5) | 105 (5) | 0.150 |
| 24-h Ambulatory SBP (mmHg) | 13 | 155 (6) | 152 (7) | 0.512 |
| 24-h Ambulatory DBP (mmHg) | 13 | 96 (4) | 93 (5) | 0.353 |
| HR (bpm) | 13 | 74 (3) | 69 (4) | 0.074 |
| **HRV** | | | | |
| Total power ($msec^2$) | 11 | 671.3 (124.2) | 1294.2 (323.3) | 0.029 |
| LF power ($msec^2$) | 11 | 224.0 (65.2) | 320.9 (76.7) | 0.231 |
| HF power ($msec^2$) | 11 | 63.7 (16.8) | 122.7 (38.6) | 0.055 |
| LF:HF ratio | 11 | 4.8 (1.0) | 4.8 (1.2) | 0.961 |

N represents the number of patients analyzed, numbers represent means (SE), p-values are obtained from paired $t$-tests. MSNA = muscle sympathetic nerve activity, BRS = baroreflex sensitivity, cBRS = cardiac BRS, xBRS = cross-correlation BRS, sBRS = sympathetic BRS, $T_{50}$ = diastolic BP at which 50% of the heartbeats is associated with a burst, CPT = cold pressor test, MAP = mean arterial pressure, HR = heart rate, BP = blood pressure, SBP = systolic BP, DBP = diastolic BP, HRV = heart rate variability, LF = low frequency, HF = high frequency.

peroneal nerve, which is one of the most technically challenging but reliable methods to assess overall sympathetic drive [24–26]. Although integrated MSNA decreased in 6 out of 10 patients (60%) following EVBA and raw MSNA decreased in 8 out of 10 patients (80%), mean MSNA changes were not significant.

Several reasons may explain the lack of a significant effect on MSNA in the present study. In the first place, this is a proof-of-principle study in a small group of patients. Secondly, the proportion of BP responders to EVBA implantation was small and mean BP changes were modest (office systolic BP decreased by 14 mmHg [95% CI 2–27] and 24-h ambulatory BP did not significantly decrease [-3 mmHg, 95% CI -7–13]). This may explain why a significant effect on MSNA was not observed. Also, calcium channel blockers that were prescribed as escape medication could have attenuated the effect, as MSNA initially increases when calcium

channel blockers are initiated [10]. Finally, the absence of a positive correlation between change in MSNA and change in BP (which was present in the studies investigating BAT [3, 7]) implies that the observed decrease in BP may not be mediated (solely) by sympathetic inhibition, but by another, currently unknown mechanism. Given the large within-patient variability of the MSNA measurement and the limited number of patients studied, these correlations and conclusions should be interpreted with caution.

Since the MobiusHD implant changes the geometric properties of the carotid sinus and is designed to modulate baroreflex pathways, an important finding of our study is that baroreflex sensitivity and cardiovascular responses to a sympathetic stimulus did not change after EVBA, indicating that cardiovascular compensatory mechanisms were preserved. The trend to decreased HR after MobiusHD implantation together with an increase in HRV predominantly in the high frequency spectrum indicates that EVBA may modulate the parasympathetic nervous system. A decrease in HR has also been observed in hypertensive patients treated with BAT: HR decreased by 4.6 bpm (95% CI 1.6–7.6) with the Rheos system [3] and by 3.6 bpm (95% CI 1.5–5.7) with the Barostim *neo* [7]. The increase in HRV during BAT, however, was not significant [3]. The effects of antihypertensive treatment on HRV have also been studied in patients treated with renal denervation [27]. After renal denervation, the LF/HF ratio in response to renal nerve stimulation decreased significantly due to a decrease in the LF component (reflecting sympathetic tone) and an increase in the HF component (reflecting parasympathetic tone). The trend to a decrease in HR in the present study contrasts our observations from the first-in-human study [8] in which HR did not change following EVBA. The difference might be explained, at least in part, by the influence of antihypertensive drugs which were temporarily discontinued in this proof-of-principle study.

The moderate BP response in the present study compared to the first-in-human study is remarkable. Mean office systolic BP in the present study decreased by 14 mmHg (95% CI 2–27) compared to 24 mmHg (95%CI 12–35) in the previous study. Also, the proportion of patients with a clinically relevant BP response (defined as at least 10 mmHg decrease in office systolic BP, at least 5 mmHg decrease in mean 24-h ambulatory systolic BP or at least 1 DDD reduction in antihypertensive medication without systolic BP increase) was lower compared to the first-in-human study: 53% vs. 83% at 3 months follow-up. The difference might be related to the heterogeneity of the patient population (i.e. the large variability in baseline BP) and the small sample size of this proof-of-principle study. Given the heterogeneous pathophysiology of primary hypertension and the varying contribution of sympathetic activation [28], a number of true non-responders could be expected. Accordingly, one would expect patients with high sympathetic activity to respond better to sympathetic-modulating treatment. Stratification on baseline MSNA, however, did not result in different BP responses. Patient response may also depend on the extent to which the MobiusHD increases vessel wall strain, which could differ from patient to patient, and may depend on the exact position of the MobiusHD device in the carotid sinus.

A strength of the present study is the use of microneurography, which is one of the most reliable but technically challenging methods to assess overall sympathetic activity [24–26]. Another strength is the implementation of a medication washout period which enabled us to perform baseline and 3-month measurements under the same conditions and with limited influence of antihypertensive drugs. At the same time, this may effect the generalizability of our findings, since effects were measured in the absence or in the presence of only one antihypertensive drug(s).

Limitations of this small, uncontrolled proof-of-principle study also need to be considered. These are very complex and challenging studies to perform in patients, as microneurography is time-consuming, the signal is easily affected, and in some patients the signal can 'simply' not

be obtained. Although antihypertensive medications were discontinued, drug screening in blood or urine was not performed to confirm the absence of antihypertensive drugs. Another obvious limitation was the lack of statistical power which weakens our findings and conclusions. Moreover, the associations between EVBA and all study endpoints should be interpreted with caution in the view of the fact that this is a non-randomized study in which period effects, regression to the mean, Hawthorne effects and placebo effects come into play. The unbiased effect on BP was being studied in two randomized, sham-controlled trials: the pivotal CALM-2 trial (Controlling And Lowering blood pressure with the MobiusHD) studying the effect of EVBA in patients with resistant hypertension on a confirmed stable antihypertensive medication regimen; and CALM-START (Controlling And Lowering blood pressure with the MobiusHD–STudying effects in A Randomized Trial), studying the effect of EVBA in patients in absence of antihypertensive medication. Due to difficulties in patient inclusion, these studies currently stopped enrolling. A final limitation of the present study is that sympathetic activity was only measured by MSNA and not by other direct or indirect measures such as norepinephrine spill-over or peripheral vascular resistance. We also did not measure volume changes and changes in cardiac output which can affect BP and might explain the absence of correlation between changes in sympathetic activity and BP.

In conclusion, in this proof-of-principle study which lacks the power to draw firm inferences, EVBA did not significantly effect sympathetic activity in patients with resistant hypertension. The observation that EVBA did not impair baroreflex sensitivity or cardiovascular responses to a sympathetic stimulus suggests that BP compensatory mechanisms are not undermined by MobiusHD implantation which is perhaps the most important finding of this study.

## Supporting information

**S1 Checklist. TREND statement checklist.**
(DOC)

**S1 File. Detailed description of study in- and exclusion criteria.**
(PDF)

**S2 File. Extensive summary study protocol.**
(DOCX)

**S3 File. Extensive summary sub-study protocol.**
(DOCX)

**S1 Data.**
(XLSX)

**S1 Fig. Schematic overview of the measurement setting.** BP = blood pressure, MSNA = muscle sympathetic nerve activity, CPT = cold pressor test.
(TIF)

**S2 Fig. MSNA recording of one of the participants (patient 8, baseline).** Heart rate (I), beat-to-beat blood pressure (II), integrated MSNA (III), filtered MSNA (IV), and pressure curve of the mouth pressure applied during the Valsalva manoeuver (V). The part between the vertical dashed lines represents early phase II of the Valsalva manoeuver.
(TIF)

**S1 Table. Number of patients excluded per outcome variable and reasons for exclusion.** N represents the number of patients excluded from the 13 patients who underwent 3-month

follow-up measurements. BRS = baroreflex sensitivity, cBRS = cardiac BRS, xBRS = cross-correlation BRS, sBRS = sympathetic BRS, $T_{50}$ = diastolic BP at which 50% of the heartbeats is associated with a burst, CPT = cold pressor test, MAP = mean arterial pressure, HR = heart rate, NA = not applicable, BP = blood pressure, SBP = systolic BP, DBP = diastolic BP, HRV = heart rate variability, LF = low frequency, HF = high frequency.
(PDF)

**S2 Table. Individual patient characteristics, MSNA and BP at baseline, and individual changes after 3 months.** DM 2 = diabetes mellitus type 2, CVD = history of cardiovascular disease, RDN = history of renal denervation, mn b2b = mean beat-to-beat, f = female, m = male, A = ACE-inhibitor or angiotensin-II-receptor blocker, C = calcium antagonist, D = diuretic, MRA = mineralocorticoid receptor antagonist, α = alpha blocker, β = beta blocker, R-i = direct renin inhibitor.
(PDF)

**S3 Table. Overview of the predefined safety outcomes that occured within 3 months after implantation.** *Only those related or possibly related to the device or procedure or with relatedness unknown. ˟Decrease in eGFR of >10%.
(PDF)

## Acknowledgments

We gratefully acknowledge all CALM-DIEM sub-study participants for their commitment and patience during the MSNA measurements. We would also like to thank Leonard van Schelven and Ernest Bošković for their technical support of the MSNA measurements.

## Author Contributions

**Conceptualization:** Monique E. A. M. van Kleef, Wilko Spiering.

**Formal analysis:** Monique E. A. M. van Kleef, Karsten Heusser, André Diedrich.

**Funding acquisition:** Wilko Spiering.

**Investigation:** Monique E. A. M. van Kleef, P. Liam Oey.

**Methodology:** Monique E. A. M. van Kleef, Karsten Heusser, André Diedrich, Wilko Spiering.

**Project administration:** Monique E. A. M. van Kleef, Wilko Spiering.

**Resources:** Peter J. Blankestijn.

**Software:** Karsten Heusser, André Diedrich.

**Supervision:** Wilko Spiering.

**Visualization:** Monique E. A. M. van Kleef.

**Writing – original draft:** Monique E. A. M. van Kleef, Karsten Heusser, André Diedrich, Wilko Spiering.

**Writing – review & editing:** Monique E. A. M. van Kleef, Karsten Heusser, André Diedrich, P. Liam Oey, Jens Tank, Jens Jordan, Peter J. Blankestijn, Bryan Williams, Wilko Spiering.

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
