## [Decision Letter · Decision Letter 0]

31 Mar 2021

PONE-D-20-24916

Endovascular baroreflex amplification and the effect on sympathetic nerve activity in patients with resistant hypertension: a proof-of-principle study

PLOS ONE

Dear Dr. Spiering,

Thank you for submitting your manuscript to PLOS ONE. After careful consideration, we feel that it has merit but does not fully meet PLOS ONE’s publication criteria as it currently stands. Therefore, we invite you to submit a revised version of the manuscript that addresses the points raised during the review process.

We look forward to receiving your revised manuscript.

Kind regards,

Michiel Voskuil, MD, PhD

Academic Editor

PLOS ONE

Journal Requirements:

3. Thank you for submitting your clinical trial to PLOS ONE and for providing the name of the registry and the registration number. The information in the registry entry suggests that your trial was registered after patient recruitment began. PLOS ONE strongly encourages authors to register all trials before recruiting the first participant in a study.

i) your reasons for your delay in registering this study (after enrolment of participants started);

ii) confirmation that all related trials are registered by stating: “The authors confirm that all ongoing and related trials for this drug/intervention are registered”.

4. Please include captions for ALL your Supporting Information files at the end of your manuscript, and update any in-text citations to match accordingly. Please see our Supporting Information guidelines for more information: http://journals.plos.org/plosone/s/supporting-information.

[Monique E.A.M. van Kleef was indirectly paid from a research grant by Vascular Dynamics, Inc. Karsten Heusser, P. Liam Oey, Jens Tank, André Diedrich and Peter J. Blankestijn report no conflict of interest. Jens Jordan served as consultant for Novartis, Boehringer-Ingelheim, Sanofi, Orexigen, Riemser, Vivus, and is cofounder of Eternygen GmbH. Bryan Williams has received honoraria for consultancy from Vascular Dynamics Inc. W. Spiering is a consultant for Vascular Dynamics and has received a research grant from Vascular Dynamics.].

Reviewers' comments:

Reviewer's Responses to Questions

**Comments to the Author**

1. Is the manuscript technically sound, and do the data support the conclusions?

Reviewer #1: Yes

Reviewer #2: Yes

2. Has the statistical analysis been performed appropriately and rigorously? 

Reviewer #1: Yes

Reviewer #2: Yes

3. Have the authors made all data underlying the findings in their manuscript fully available?

Reviewer #1: Yes

Reviewer #2: Yes

4. Is the manuscript presented in an intelligible fashion and written in standard English?

Reviewer #1: Yes

Reviewer #2: Yes

5. Review Comments to the Author

Reviewer #1: A proof-of-principle study (n=14/10 evaluable) was conducted to explore potential mechanisms for lowering blood pressure. The effects of endovascular baroreflex amplification (EVBA) on muscle sympathetic nerve activity (MSNA) and baroreceptor sensitivity (BRS) were investigated. The primary outcome was change in MSNA. Secondary outcomes were change in baroreflex sensitivity (BRS), cardiovascular responses to a sympathetic stimulus, BP, heart rate (HR) and heart rate variability (HRV). The results are primarily descriptive.

Minor revisions:

1- Line 223: Report occurrences of adverse events with both frequencies and percentages. To provide additional clarity, consider providing a table which summarizes the adverse events.

2- Provide percentages which correspond to the number of patients out of the total evaluated. For example, consider Line 257, indicate that 6 of 10 is 60%.

3- Line 262: Explain how the following statement can be justified statistically. “The trend to decrease in MSNA could not be explained by change in quality of the MSNA measurement, since MSNA signal-to-noise ratio at baseline and 3 months post-implantation were similar.”

Reviewer #2: Van Kleef and colleagues investigated the effects of a novel blood pressure lowering device (Mobius HD) on muscle sympathetic nerve activity (MSNA) and baroreceptor sensitivity (BRS). Mobius HD is a self-expanding nitinol implant that is delivered percutaneously to the internal carotid sinus using an intravascular delivery system. The objective of the implant is to amplify the baroreceptor signaling resulting in reduction of blood pressure through vasodilation. The manuscript is well written and easy to follow and provides us with interesting data. The authors should be congratulated on their work. However, I have some concerns regarding several aspects of the paper that require revisions.

1. This is an ancillary investigation of the CALM-DIEM study. The investigators published their data of the first-in-human trial of this implant in patients with resistant hypertension (Spiering et al. Lancet, 2017). This device was associated with a reduction in 24-hour ambulatory systolic blood pressure of 21/12 mmHg at six months after implantation.

2. The number of patients included in the study seems to be insufficient to draw firm conclusions.

3. The authors report that they did not find a correlation between change in MSNA and BP after Mobius HD implantation. Current study needs to be discussed in light of publications such as Hoogerwaard et al (Clin Res Cardiol, 2019). They reported on the effects of high frequency electrical stimulation of renal nerves on heart rate variability, a non-invasive tool for autonomic nervous system function. Before RDN, high frequency electrical stimulation induced changes in heart rate variability suggesting alteration in autonomic balance with an increased sympathetic autonomic balance tone. Whereas, after RDN, the changes in heart rate variability reflected a lower sympathetic tone and greater parasympathetic tone after RDN.

4. Continuation and discontinuation of beta blockers during follow up need to be reported in conjunction with the statement of changes in MSNA and BRS. Hoogerwaard et al. (Clin Res Cardiol, 2019), reported that changes in heart rate variability were 1.5 times greater in patients off-beta-blockade, and no longer significant in patients on-beta-blockade and in a small group of patients with diabetes mellitus. Did you correct for these potential confounders?

5. While the results of this study are intriguing, it is important to underline that this is a sub-study of a previously published study that lacks power with the limited of patients in the current study weakening the findings and conclusions. The data and the findings are interesting and mostly hypothesis generating.

6. PLOS authors have the option to publish the peer review history of their article (what does this mean?). If published, this will include your full peer review and any attached files.

Reviewer #1: No

Reviewer #2: No

---

## [Author Response · Author response to Decision Letter 0]

3 Jun 2021

Dear Dr. Voskuil, 

We are pleased to submit the revised version of our manuscript ”Endovascular baroreflex amplification and the effect on sympathetic nerve activity in patients with resistant hypertension: a proof-of-principle study” after the valuable comments raised by the reviewers. This letter contains a point-by-point response to the issues raised by them.

Journal Requirements

Fig 2, S1 Fig and S2 Fig have been edited to meet the figure size requirements. 

2. Please review your reference list to ensure that it is complete and correct. 

The reference list has been reviewed and is correct. 

3. As per the journal’s editorial policy, please include in the Methods section of your paper: i) your reasons for your delay in registering this study (after enrolment of participants started); ii) confirmation that all related trials are registered by stating: “The authors confirm that all ongoing and related trials for this drug/intervention are registered”.

We added the reasons for our delay and confirmation that all trials are registered to the Methods section, lines 98-101: “Because of its small size and the enrolment of patients in a single center only, the present sub-study was initially not registered with the main study. The sub-study was added after enrolment of the participants started. We confirm that all ongoing and related trials for this intervention are registered.”

4. Please include captions for all your Supporting Information files at the end of your manuscript, and update any in-text citations to match accordingly.

Captions for all Supporting Information files are included at the end of the manuscript. 

5. Thank you for stating the following “Monique E.A.M. van Kleef … Vascular Dynamics” in the Competing Interests section. Please confirm that this does not alter your adherence to all PLOS ONE policies on sharing data and materials, by including the following statement: "This does not alter our adherence to PLOS ONE policies on sharing data and materials.” 

We confirm that the competing interests stated do not alter our adherence to PLOS ONE policies on sharing data and materials. You may add the statement "This does not alter our adherence to PLOS ONE policies on sharing data and materials” to our Competing Interests section. 

Reviewer 1

1) Line 223: Report occurrences of adverse events with both frequencies and percentages. To provide additional clarity, consider providing a table which summarizes the adverse events.

In line 227-228 we added “and percentage of the total study population”. A table summarizing the adverse events is added to the Supporting Information in S3 Table. 

2) Provide percentages which correspond to the number of patients out of the total evaluated. For example, consider Line 257, indicate that 6 of 10 is 60%.

The percentages corresponding to the number of patients evaluated were added to: 

- Line 261: (60%) 

- Line 263: (80%)

- Line 299: (69%)

- Line 305: (7%)

- Line 312: (14%)

- Line 317: , 21%

- Line 317: , 14%

- Line 317: , 7%

- Line 318: , 7%

- Line 318: , 7%

- Line 319: , 7%

- Line 331: (60%)

- Line 331: (80%)

3) Line 262: Explain how the following statement can be justified statistically. “The trend to decrease in MSNA could not be explained by change in quality of the MSNA measurement, since MSNA signal-to-noise ratio at baseline and 3 months post-implantation were similar.”

Thank you for this comment, we realize that this statement needs some more explanation. In a situation with a relatively high level of noise (a low signal-to-noise ratio) it is more difficult for the software to distinguish between peaks caused by noise and peaks caused by the mechanism we are interested in: sympathetic nerve activity. Theoretically the level of noise could be lower the second measurement due to training or habituation. Therefore, we only included measurements with a signal-to-noise ratio of >2 [1] and evaluated whether the mean signal-to-noise ratio at baseline differed from the mean signal-to-noise ratio at 3 months. Since this is an ‘exploratory’ analysis in a proof-of-principle study that already lacks power to show a significant change in the primary endpoint (MSNA), we believe there is no additional value in statistically testing the difference in mean measurement quality. 

Reviewer 2

1) This is an ancillary investigation of the CALM-DIEM study. The investigators published their data of the first-in-human trial of this implant in patients with resistant hypertension (Spiering et al. Lancet, 2017). This device was associated with a reduction in 24-hour ambulatory systolic blood pressure of 21/12 mmHg at six months after implantation.

We published the 6-month data of the European cohort of the first-in-human study (CALM-FIM: Controlling And Lowering blood pressure with MobiusHD – First In Man, NCT01831895) in 2017. Publication of the 3-year follow-up data of both the American and European cohorts of the CALM-FIM study is expected soon. In the present manuscript we present a sub-study of a second trial investigating the effects in an uncontrolled design: CALM-DIEM (Controlling And Lowering blood pressure with MobiusHD – Defining Efficacy Markers, NCT02827032). The CALM-DIEM sub-study was specifically designed to investigate the mechanism-of-action of EVBA. 

2) The number of patients included in the study seems to be insufficient to draw firm conclusions.

Together with the uncontrolled design of the study, the small number of patients is indeed one of the major limitations of this work. We have based the population size on a sample size calculation assuming the effect size to be greater and the variation to be smaller than we actually observed. Therefore, one might conclude that this study was underpowered. However, also the small proportion of blood pressure responders and the modest blood pressure changes observed in this study, may provide an explanation for the lack of a significant effect on sympathetic nerve activity. Indeed, all these factors impede us to draw definite conclusions. We discuss these limitations in lines 333-338 and 391-393 of the Discussion. 

3) The authors report that they did not find a correlation between change in MSNA and BP after Mobius HD implantation. Current study needs to be discussed in light of publications such as Hoogerwaard et al (Clin Res Cardiol, 2019). They reported on the effects of high frequency electrical stimulation of renal nerves on heart rate variability, a non-invasive tool for autonomic nervous system function. Before RDN, high frequency electrical stimulation induced changes in heart rate variability suggesting alteration in autonomic balance with an increased sympathetic autonomic balance tone. Whereas, after RDN, the changes in heart rate variability reflected a lower sympathetic tone and greater parasympathetic tone after RDN.

Thank you for referring to this interesting article providing more insight into the mechanism by which renal denervation reduces the exaggerated blood pressure response to sympathetic stimulation. The study by Hoogerwaard et al confirms that the renal nerve stimulation-induced effect on blood pressure is reduced after renal denervation and shows that this can be explained by a change in the sympathovagal balance toward a higher parasympathetic tone. We incorporated this interesting finding into our discussion “The effects of antihypertensive treatment on HRV have also been studied in patients treated with renal denervation. After renal denervation, the LF/HF ratio in response to renal nerve stimulation decreased significantly due to a decrease in the LF component (reflecting sympathetic tone) and an increase in the HF component (reflecting parasympathetic tone).” (lines 355-359). 

In our proof-of-principle study we also investigated whether the BP response to a sympathetic stimulus (the cold pressor test) changed after treatment with EVBA. We observed that BP response to the cold pressor test remained unchanged after EVBA. Unfortunately, we did not evaluate changes in HRV evoked by the cold pressor test. 

It is not quite possible to make a direct comparison between the change in BP response and change in HRV reflecting a lower sympathetic tone in the study by Hoogerwaard et al, and the correlation between change in MSNA reflecting sympathetic tone and change BP in our study. First, Hoogerwaard et al investigated the change in the renal nerve stimulation-induced effect on BP while we evaluated the change in non-induced BP after treatment. Second, Hoogerwaard et al relate the mean change in HRV to the mean change in the renal nerve stimulation-induced effect on BP, while we correlated individual changes in MSNA to individual changes in BP. 

4) Continuation and discontinuation of beta blockers during follow up need to be reported in conjunction with the statement of changes in MSNA and BRS. Hoogerwaard et al. (Clin Res Cardiol, 2019), reported that changes in heart rate variability were 1.5 times greater in patients off-beta-blockade, and no longer significant in patients on-beta-blockade and in a small group of patients with diabetes mellitus. Did you correct for these potential confounders?

This is an interesting statement since beta blockers could blunt the effect on sympathetic activity. Therefore, beta blocker use could be an effect modifier in the association between EVBA and sympathetic activity. There are two reasons why we could not stratify our analyses into patients with or without beta blockers. Per protocol patients were instructed to taper and discontinue beta blockers in the four to two weeks before study measurements were performed. The length of this washout period ensures that sympathetic and cardiovascular measurements were not affected by beta blockers. Furthermore, there was only one patient with a beta blocker included in the study. Since this patient was excluded from the analysis due to missing data at 3 months, there were no patients previously using beta blockers to be analyzed. 

5) While the results of this study are intriguing, it is important to underline that this is a sub-study of a previously published study that lacks power with the limited of patients in the current study weakening the findings and conclusions. The data and the findings are interesting and mostly hypothesis generating.

This is indeed a sub-study of a previously registered study, the CALM DIEM study (Controlling And Lowering blood pressure with MobiusHD – Defining Efficacy Markers, NCT02827032). We would like to emphasize that this is not a post-hoc analysis of a previously published study. The endpoints of the present study were determined before the sub-study enrolled any patients. The study that has been published was our first-in-man study, CALM FIM (Controlling And Lowering blood pressure with MobiusHD – First In Man, NCT01831895). 

We agree that the small sample size precludes us from drawing firm conclusions, and that the findings of this study are mostly hypothesis generating. To emphasize this study limitation we added two lines to our discussion: “which weakens our findings and conclusions” (line 392-393) and “which lacks the power to draw firm inferences” (line 407). 

We hope that this rebuttal letter clarifies our thoughts and the decisions we made. We believe the revisions have improved the transparency of our results and the quality of the manuscript. Therefore, we hope you will reconsider publication of our paper in PLOS ONE. 

We are looking forward to your response. 

Yours sincerely, 

On behalf of the co-authors,

Wilko Spiering, MD PhD

---

## [Decision Letter · Decision Letter 1]

28 Oct 2021

Endovascular baroreflex amplification and the effect on sympathetic nerve activity in patients with resistant hypertension: a proof-of-principle study

PONE-D-20-24916R1

Dear Dr. Spiering,

We’re pleased to inform you that your manuscript has been judged scientifically suitable for publication and will be formally accepted for publication once it meets all outstanding technical requirements.

Kind regards,

Michiel Voskuil, MD, PhD

Academic Editor

PLOS ONE

Reviewers' comments:

Reviewer's Responses to Questions

**Comments to the Author**

1. If the authors have adequately addressed your comments raised in a previous round of review and you feel that this manuscript is now acceptable for publication, you may indicate that here to bypass the “Comments to the Author” section, enter your conflict of interest statement in the “Confidential to Editor” section, and submit your "Accept" recommendation.

Reviewer #1: All comments have been addressed

Reviewer #2: All comments have been addressed

2. Is the manuscript technically sound, and do the data support the conclusions?

Reviewer #1: (No Response)

Reviewer #2: Yes

3. Has the statistical analysis been performed appropriately and rigorously? 

Reviewer #1: (No Response)

Reviewer #2: Yes

4. Have the authors made all data underlying the findings in their manuscript fully available?

Reviewer #1: (No Response)

Reviewer #2: Yes

5. Is the manuscript presented in an intelligible fashion and written in standard English?

Reviewer #1: (No Response)

Reviewer #2: Yes

6. Review Comments to the Author

Reviewer #1: (No Response)

Reviewer #2: The authors adequately addressed the issues raised by the reviewers. The manuscript improved significantly after the revisions. No further comments.

7. PLOS authors have the option to publish the peer review history of their article (what does this mean?). If published, this will include your full peer review and any attached files.

Reviewer #1: No

Reviewer #2: No

---

## [Editor Report · Acceptance letter]

8 Nov 2021

PONE-D-20-24916R1 

Endovascular baroreflex amplification and the effect on sympathetic nerve activity in patients with resistant hypertension: a proof-of-principle study 

Dear Dr. Spiering:

I'm pleased to inform you that your manuscript has been deemed suitable for publication in PLOS ONE. Congratulations! Your manuscript is now with our production department. 

Kind regards, 

on behalf of

Dr. Michiel Voskuil 

Academic Editor

PLOS ONE